# Motivational Climate towards the Practice of Physical Activity, Self-Concept, and Healthy Factors in the School Environment

**Manuel Castro-Sánchez** [1,]*, **Félix Zurita-Ortega** [1], **Eduardo García-Marmol** [2] and **Ramón Chacón-Cuberos** [3]

1　Department of Didactics of Musical, Plastic and Corporal Expression, University of Granada, 18071 Granada, Spain; felixzo@ugr.es
2　Department of Physical Education and Sports, University of Granada, 18071 Granada, Spain; eduardogarcia@ugr.es
3　Department of Research and Diagnosis Methods in Education, University of Granada, 18071 Granada, Spain; rchacon@ugr.es
*　Correspondence: manuelcs@ugr.es; Tel.: +34-958-248-949

**Abstract:** The objective of the present study was to define and contrast an explanatory model relating the motivational climate, body mass index, and adherence to a Mediterranean diet with the self-concept of school children. A further objective was to analyze the existing relationships between the variables included in the developed model according to sex, using a multi-group structural equation analysis. In the study, a total of 734 school children, of both sexes, reported their perceived motivational climate towards sport, body mass index, adherence to a Mediterranean diet, and self-concept. These children were aged between 10 and 12 years old and attended public school in the province of Granada (Spain). The multi-group structural equation model developed demonstrated an excellent fit to the empirical data ($\chi2$ = 228.179; DF = 40; $p < 0.001$; CFI = 0.965; NFI = 0.958; IFI = 0.968; RMSEA = 0.048). The findings identified a direct negative relationship between the ego climate and the task climate. Furthermore, an inverse relationship was found between the task climate and body mass index, and a direct relationship was found between the ego climate and body mass index. Meanwhile, Mediterranean diet adherence was directly related to the task climate and negatively related to the ego climate. The main conclusions of the present study highlight the positive effects of a task-oriented motivational climate and adherence to a Mediterranean diet, with regards to body mass index. Furthermore, a task-oriented motivational climate and a lower body mass index are related to a more positive self-concept.

**Keywords:** motivational climate; Mediterranean diet; body mass index; self-concept

---

## 1. Introduction

Motivation is of their motivations, choosing to engage in certain behaviors and avoid engaging in others [1]. In the context of physical activity, the study of motivation serves to improve the understanding of the multitude of psychological and behavioral factors that relate to its engagement [2]. Motivation is understood to result from the interaction between diverse variables (cognitive, biological, social, and emotional) which determine the decision to engage, the intensity of engagement, and the performance of a concrete activity [3].

The study of different motivation aspects related to sport and physical activity has been framed by two key theories: The self-determination theory [4] and the achievement goal theory [5]. These theories are currently considered to be highly relevant for the analysis of motivation in association with the

practice of physical activity and sport [6]. Both lend special interest to the social factors that influence individual decisions to engage in physical activity and are based on the idea that when individuals are confronted with different situations, their motivations are directed either towards the desire for successful execution or towards the demonstration of personal ability [7].

The interpretation of success and failure is personal and depends on whether the individual is oriented towards their ego (they desire to demonstrate their superiority over others), or towards the task (they desire personal improvement) [8]. The orientation assumed by an individual is influenced by several dispositional factors, all acting at the same time. These include the individual's tendency towards one orientation over another and environmental influences, such as the motivational climate, which promotes the emphasis of certain processes over others in the context of physical activity and healthy habits [9].

Motivation is a crucial element of any learning process. This is true as much in the context of sport as it is in education, given that individuals are more effective in achieving difficult learning outcomes when they set defined objectives and work towards the achievement of personal goals. This, in turn, translates into greater motivation [10]. Accordingly, the motivational climate constitutes the social environment and is defined by implicit and explicit processes that predispose individuals to the attainment of success [11]. Depending on how these implicit and explicit processes are used (design of tasks, use of rewards, working with others, and evaluation), motivational climates oriented towards mastery (task) or performance (ego) are developed [12].

Several studies have been conducted by the scientific community that identify cognitive, physical, and social benefits of regularly engaging in physical activity [13,14]. Engagement in physical activity is incrementally related to a reduced risk of suffering illnesses associated with obesity or being overweight, improvements in cardiovascular capacity, improved social relations, and diverse mental benefits [15].

In association with engagement in sport, dietary patterns are a key factor with a strong influence on the state of the health of school children [16]. Currently, the dietary habits of school children are often characterized by an elevated consumption of processed sugars and saturated fats, alongside a deficiency of fruit, vegetables, fish, and healthy fats [17]. In contrast, the Mediterranean diet, hailing from countries such as Italy and Spain, exerts a positive impact on the health of individuals [18]. This type of diet is characterized by a high content of legumes, fruits, vegetables, fish, and natural antioxidants such as olive oil, as well as moderate consumption of fats, eggs, and meat [19]. Various studies have shown that high adherence to this dietary model has positive repercussions on the health of defined populations [20,21].

Self-concept represents a key psychological factor in the development of individuals during the school stage. This factor is defined as the perception held by an individual about themselves and relates to a number of dimensions, namely family, emotional, academic, physical, and social [22]. Previous studies, such as those conducted by Turner, Shattuck, Finkelhor, and Hamby [23], have remarked on the important role of self-concept in the configuration of the personality of an individual. It is considered to be a crucial factor, contributing to the wellbeing of the individual. Self-concept is, in turn, influenced by a multitude of factors, including motivation, degree of obesity or overweight, social relationships, and academic performance, amongst others [24].

This research aims to analyze the constructs mentioned in the previous paragraphs, due to the importance of these factors in the psychosocial health of school children. These subjects in school-age children have certain characteristics that distinguish them from those in adolescents and adults, due to a change from childhood to preadolescence occurring at this age [25]. During this stage, the physical development of school children has reached maturity and harmony, preparing the body for the pubertal developments characteristic of adolescence [26]. In addition, there are several changes in cognition that occur, such as the development of the capacity for abstraction and the development of more complex logical processes [27]. Regarding the emotional field, school children begin to suffer from affective alterations due to the onset of sexuality, as well as changes in their self-concept, their emotional

perception, and the way they socialize with others. At a social level, the main socializing agent ceases being the family, to the detriment of the peer groups which obtain greater autonomy [28]. Due to these characteristics, the population of school children from 10 to 12 years old is coined as a focus of interest in the present research.

There are several investigations in which the motivational climate has been analyzed in relation to a multitude of factors, such as emotional intelligence or stress, and in various populations, such as adolescents, university students, or athletes [13–21]. Nevertheless, the present research uses a theoretical model which analyzes the possible association between motivational factors, self-concept, and variables indicating health status (such as adherence to the Mediterranean diet and body mass index) in a sample of school children from 10 to 12 years old. Therefore, the main novelty of this study lies in the analysis of psychological and health factors in a school population through a structural equation model.

In relation to the literature reviewed, this research proposed the following objectives:

- To contrast an explanatory model incorporating the motivational climate, body mass index, Mediterranean diet adherence, and self-concept.
- To analyze, using multi-group structural equation analysis, the pertinent relationships between the motivational climate, body mass index, Mediterranean diet adherence, and self-concept, according to the sex of school children (male/female).

## 2. Materials and Methods

### 2.1. Subjects and Design

The present study employed a descriptive cross-sectional design and included a sample of 734 school children of both sexes (45.2% male and 54.8% female). Participants were aged between 10 and 12 years (M = 10.88 years; SD = 0.69) and were enrolled in the fifth or sixth year of their primary education in the city of Granada at the time of the data collection. Convenience sampling was used to recruit the participants, with only school children in the third cycle of their primary education being invited to participate. The sample was recruited from 11 public schools in the capital city of Granada, with the participation of all schools being entirely voluntary. Individuals were monitored during the collection of the data to ensure that participants only completed the measures once; in this way, the duplication of data was avoided.

### 2.2. Measures

The motivational climate (PMCSQ-2) scale was adapted from the original version, developed by Newton, Duda, and Yin [29], and translated into Spanish by González-Cutre et al. [30]. This questionnaire is composed of 33 items which are pointed by the means of a five-point Likert scale, ranging from 1 to 5 (totally disagree/totally agree). This scale measures the responses according to two categories: The first is the task-oriented climate, which consists of the categories of an important role, effort/improvement, and cooperative learning. The second is the ego-oriented climate, which consists of the categories of unequal recognition, the punishment of mistakes, and rivalry between group members. The reliability of the instrument in its translated Spanish form has been previously reported by González-Cutre et al. [30] using the Cronbach alpha coefficient. The values obtained were $\alpha = 0.90$ for the ego-oriented climate dimension and $\alpha = 0.84$ for the task-oriented climate dimension.

Meanwhile, the Mediterranean diet quality index (KIDMED) questionnaire was developed by Serra-Majem et al. [31]. The KIDMED is composed of 16 items which relate to different aspects of the Mediterranean diet and are reported dichotomously (Yes and No). Items 5, 11, 13, and 15 are negatively framed, whilst all other items are positively framed. The responses for all of the items are summated, in order to get a possible overall score of between $-4$ and 12 points. This overall score is then classified according to three categories: a) Low-quality diet ($\leq$ 3 points), b) Needs improvement

(4–7), and c) Optimal diet ($\geq$ 8). The present study found the KIDMED to be reliable, with a Cronbach alpha of $\alpha$ = 0.92.

Body mass index (BMI) was calculated from the participant's weight (in kilograms) divided by the square of their height (in meters). The weight of the participants was measured using weighing scales ("Seca 811") and a stadiometer ("Seca 213").

The self-concept form-5 (AF-5) questionnaire, developed by García and Musitu [32] and based on the theoretical model of Shavelson et al. [33], was used. The AF-5 is composed of 30 items rated on a five-point Likert scale, where 1 corresponds to "never" and 5 corresponds to "always". The instrument is designed to measure responses according to five dimensions of self-concept, including academic self-concept, social self-concept, emotional self-concept, family self-concept, and physical self-concept. In research conducted by García and Musitu [32], the internal consistency of the instrument was established (determined through the Cronbach alpha coefficient) as $\alpha$ = 0.81. A similar value was identified in the present study ($\alpha$ = 0.83).

### 2.3. Procedure

Firstly, the participation of a convenience sample of primary schools meeting the inclusion criteria in the capital city of Granada was solicited through the Faculty of Education Science at the University of Granada. The management at each school was informed about the nature of the research and agreement of the school was requested for the participation of their pupils. Secondly, given that eligible participants were under 18 years of age, information packs were sent to the parents and legal guardians of the school children, requesting informed consent for their child's participation.

The anonymity of all information provided during the data collection was guaranteed at all times and participants were informed that the data would only be used for research purposes. Members of the research team responsible for administering the questionnaires were present at all times throughout the data collection, in order to resolve any doubts. Nevertheless, no issues or abnormalities were reported. Finally, the teaching staff, administrative staff, and all of the collaborators who assisted in the production of the study were thanked for their assistance and told that they would be sent a summary of the relevant outcomes shortly after the study ended. This summary would also respect the confidentiality of individuals.

The initial sample consisted of a total of 786 school children. A total of 52 questionnaires were excluded due to incorrect or incomplete completion of the data collected. Nevertheless, it was detected that a total of 52 questionnaires contained unfilled data items; for this reason, the data obtained from these subjects was discarded because it was not considered valid. The present study followed the principles relating to research projects, as stated in the Declaration of Helsinki (World Medical Association, 2008), for biomedical research (Law 14/2007 of the 3rd of July) and for participant confidentiality (Law 15/1999 of the 13th of December).

### 2.4. Statistical Analysis

Descriptive statistics were developed using IBM SPSS®version 22.0. The software IBM AMOS®23 was employed in order to analyze the pertinent relationships between the variables included in the path model. After developing the path model, the analysis of the relationships within the relation matrix was conducted via a multi-group analysis, grouping participants according to sex. To this end, two separate structural models were constructed, with the aim of verifying the relationships between the different construct employed as a function of the sex of the participant.

The model developed for path analysis was constructed from nine observable variables and two latent variables which described the indicators (Figure 1). Causal explications of the latent variables are made by considering the observed associations between the indicators, in combination with the reliability of the measurements. In this way, the measurement error of the observable variables is included in the model and can be directly controlled. Unidirectional arrows represent the lines of

influence between the latent and observable variables, being interpreted using regression weights. Bidirectional arrows allow the association between the latent variables to be identified.

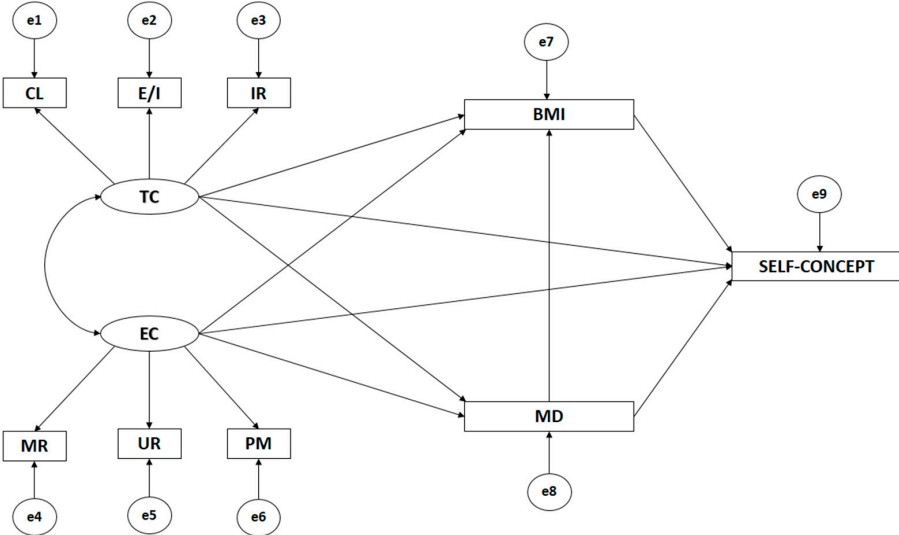

**Figure 1.** The Theoretical Model. Note: TC, task-oriented climate; E/I, effort/improvement; CL, cooperative learning; IR, important role; EC, ego-oriented climate; MR, rivalry between members; UR, unequal recognition; PM, punishment of mistakes; MD, Mediterranean diet; BMI, body mass index; SELF-CONCEPT, self-concept.

The task-oriented motivational climate (TC) and the ego-oriented motivational climate (EC) are the exogenous variables, with each being associated with three indicators. The indicators of the TC are E/I (Effort/Improvement), IR (Important Role), and CL (Cooperative Learning). In the case of the EC, the indicators are UR (Unequal Recognition), PM (Punishment of Mistakes), and MR (Rivalry between members). Adherence to a Mediterranean diet represents the endogenous variable which receives the effects of the TC and EC. Body mass index (BMI) also acts as an endogenous variable which receives the effects of the TC, EC, and MD. Finally, self-concept represents an endogenous variable, receiving the effects of the TC, EC, MD, and BMI.

The model fit was examined in order to verify the compatibility between the developed model and the empirical data obtained. The reliability of the model fit was examined according to the goodness of the fit criteria described by Marsh [34]. In the case of the chi-squared analysis, values associated with a non-significant p-value indicate a good model fit. The comparative fit index (CFI) values will be deemed to be acceptable when they are higher than 0.90 and excellent when they are higher than 0.95. The normative fit index (NFI) values should be greater than 0.90. The incremental fit index (IFI) values will be deemed acceptable when they are higher than 0.90 and excellent when they are higher than 0.95. Finally, the values for the root mean square error approximation (RMSEA) will be deemed to be excellent if they are lower than 0.05 and acceptable if they are lower than 0.08.

## 3. Results

The path model developed by examining variables measured from school children as a function of their sex demonstrates a good fit for all of the evaluation indices. Chi-squared analysis produced a significant p-value ($\chi2 = 228.179$; DF = 40; $p < 0.001$). However, it must be highlighted that this index cannot be interpreted in a standardized manner, given its susceptibility to the influence of sample size [34]. To address this, other indices of standardized fit were used, which are less sensitive to sample size. The comparative fit index (CFI) analysis obtained a value of 0.965, which describes an excellent fit. The normalized fit index (NFI) analysis produced a value of 0.958 and the incremental fit index (IFI) value was 0.968, both of which are excellent. The root mean square error approximation (RMSEA) analysis obtained an excellent value of 0.048.

Both Figure 2 and Table 1 show the regression weights for the parameters of the path model for male school children. Statistically significant associations, at the level of $p < 0.001$, can be seen between both categories of the motivational climate and their dimensions, with the relationships being positive. The association between the task-oriented climate (TC) and the ego-oriented climate (EC) was significant at the level of $p < 0.001$, being negative (r = −0.318). In this case, the regression weight shows a low to medium strength.

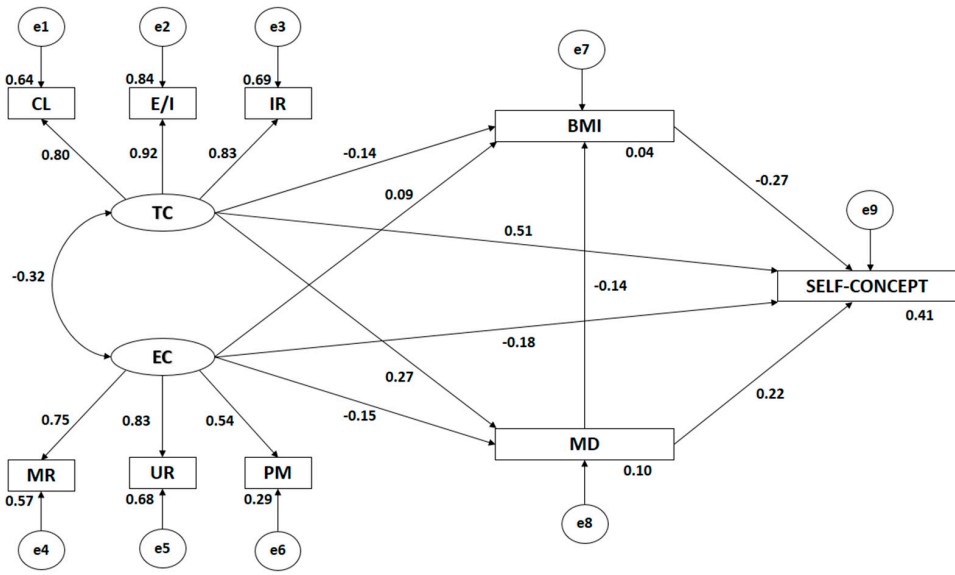

**Figure 2.** The structural equation model for the males. Note: TC, task-oriented climate; E/I, effort/improvement; CL, cooperative learning; IR, important role; EC, ego-oriented climate; MR, rivalry between members; UR, unequal recognition; PM, punishment of mistakes; MD, Mediterranean diet; BMI, body mass index; SELF-CONCEPT, self-concept.

**Table 1.** The structural model for the boys.

| Associations between Variables | | | R.W. | | | | S.R.W. |
|---|---|---|---|---|---|---|---|
| | | | Estimations | E.E. | C.R. | P | Estimations |
| MD | ← | TC | −0.182 | 0.029 | −6.274 | *** | −0.272 |
| MD | ← | EC | −0.184 | 0.057 | −3.258 | 0.019 | −0.152 |
| BMI | ← | TC | −1.887 | 0.723 | −2.609 | 0.038 | −0.140 |
| BMI | ← | EC | 0.502 | 0.253 | 1.984 | 0.047 | 0.093 |
| BMI | ← | MD | −34.737 | 11.763 | −2.953 | 0.030 | −0.138 |
| CL | ← | TC | 1.000 | – | – | *** | 0.802 |
| E/I | ← | TC | 1.187 | 0.046 | 25.900 | *** | 0.917 |
| IR | ← | TC | 1.049 | 0.044 | 23.792 | *** | 0.831 |
| MR | ← | EC | 1.000 | – | – | *** | 0.541 |
| UR | ← | EC | 0.654 | 0.055 | 11.941 | *** | 0.825 |
| PM | ← | EC | 1.222 | 0.101 | 12.124 | *** | 0.753 |
| SELF-CONCEPT | ← | TC | 0.389 | 0.030 | 13.084 | *** | 0.509 |
| SELF-CONCEPT | ← | EC | −0.193 | 0.042 | −4.599 | *** | −0.180 |
| SELF-CONCEPT | ← | MD | 3.134 | 0.756 | 4.146 | *** | 0.224 |
| SELF-CONCEPT | ← | BMI | −0.082 | 0.014 | −5.750 | *** | −0.266 |
| EC | ← | TC | −0.123 | 0.020 | −6.009 | *** | −0.318 |

Note 1: TC, task-oriented climate; E/I, effort/improvement; CL, cooperative learning; IR, important role; EC, ego-oriented climate; MR, rivalry between members; UR, unequal recognition; PM, punishment of mistakes; MD, Mediterranean diet; BMI, body mass index; SELF-CONCEPT, self-concept; Note 2: R.W., Regression Weights; S.R.W., Standardised Regression Weights; E.E., Estimation Error; C.R., Critical Ratio; Note 3: $p < 0.001$***.

In analyzing the indicators for each individual latent variable, statistically significant differences at the level of $p < 0.001$ can be seen, with all of the associations being positive. For the TC, the E/I is the indicator with which it demonstrates the greatest relation coefficient ($r = 0.917$), followed by the IR ($r = 0.831$) and CL ($r = 0.802$). For the EC, the greatest association was identified with the UR ($r = 0.825$), followed by the PM ($r = 0.753$) and MR ($r = 0.541$). These values show a high strength in all of the regression weights, except for the category rivalry between the members.

Similarly, significant associations were observed ($p < 0.001$) in the relationships between the task climate and BMI, with these being negative ($r = -0.14$) with a low strength in the regression weight. There is a significant association between the TC and MD ($p < 0.01$), with this being positive ($r = 0.27$), showing a low strength in both cases. A statistically significant difference at the level of $p < 0.001$ is also seen between the TC and self-concept, being positive ($r = 0.509$) and revealing a medium strength for the regression weights.

Analysis of the relationship between the EC and BMI identified statistically significant associations ($p < 0.01$) which are positive ($r = 0.093$). Furthermore, a relationship was found between the EC and MD ($p < 0.01$) in a negative direction ($r = -0.152$), demonstrating a low regression weight for both associations. The association between the EC and self-concept was also significant to the level of $p < 0.001$, with this relationship being negative in nature with a low strength in the regression weight ($r = -0.180$).

The MD and BMI revealed a negative association ($r = -0.138$; $p < 0.01$), with the two variables being only weakly correlated. In contrast, the relationship examined between the MD and self-concept was found to be positive, with a low strength in the regression weight ($r = 0.224$; $p < 0.001$). Finally, the relationship between BMI and self-concept is negative ($r = -0.266$), showing a significance level of $p < 0.001$ but a low strength for the regression weight.

Both Figure 3 and Table 2 show the estimated values for all of the parameters of the structural model developed from the females of the present sample of school children. Significant relationships at the level of $p < 0.001$ can be observed between both categories of the motivational climate and all of its dimensions, with these being positive. The association between the TC and EC is significant at the level of $p < 0.001$, being negative with a medium strength for the regression weight obtained ($r = -0.357$).

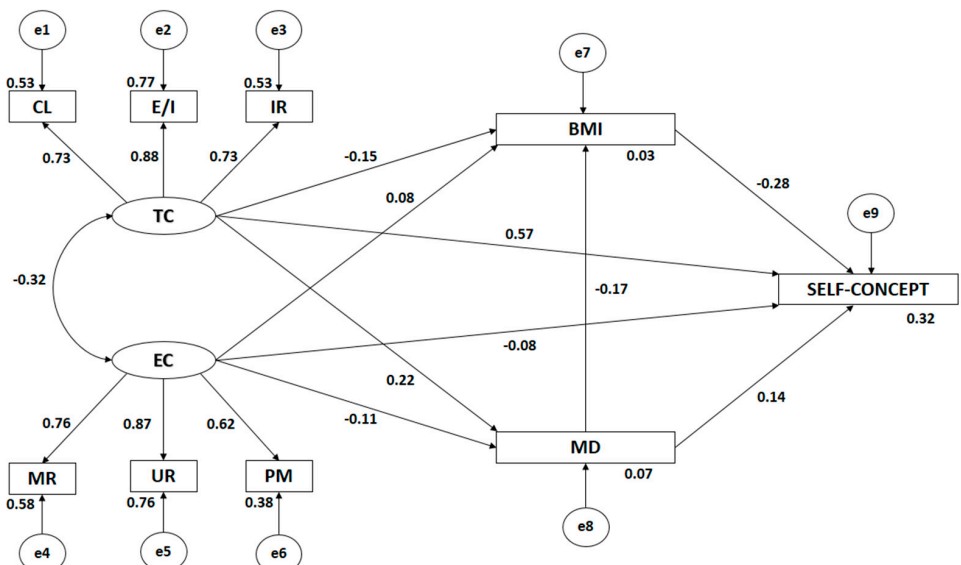

**Figure 3.** The structural equation model for the girls. Note: TC, task-oriented climate; E/I, effort/improvement; CL, cooperative learning; IR, important role; EC, ego-oriented climate; MR, rivalry between members; UR, unequal recognition; PM, punishment of mistakes; MD, Mediterranean diet; BMI, body mass index; SELF-CONCEPT, self-concept.

**Table 2.** The structural model for the females.

| Associations between Variables | | | R.W. | | | | S.R.W. |
|---|---|---|---|---|---|---|---|
| | | | Estimations | SE.E. | C.R. | P | Estimations |
| MD | ← | TC | 0.184 | 0.035 | 5.291 | *** | 0.222 |
| MD | ← | EC | −0.014 | 0.007 | −2.027 | 0.028 | −0.111 |
| BMI | ← | TC | −24.550 | 8.539 | −2.875 | 0.002 | −0.153 |
| BMI | ← | EC | 0.012 | 0.007 | 1.687 | 0.031 | 0.081 |
| BMI | ← | MD | −0.029 | 0.009 | −3.125 | *** | −0.174 |
| CL | ← | TC | 1.000 | – | – | *** | 0.726 |
| E/I | ← | TC | 1.170 | 0.056 | 20.938 | *** | 0.877 |
| IR | ← | TC | 0.951 | 0.050 | 18.884 | *** | 0.725 |
| MR | ← | EC | 1.000 | – | – | *** | 0.616 |
| UR | ← | EC | 0.564 | 0.035 | 15.899 | *** | 0.872 |
| PM | ← | EC | 1.054 | 0.065 | 16.154 | *** | 0.759 |
| SELF-CONCEPT | ← | TC | 0.465 | 0.034 | 13.603 | *** | 0.571 |
| SELF-CONCEPT | ← | EC | 0.062 | 0.029 | 2.160 | 0.014 | −0.079 |
| SELF-CONCEPT | ← | MD | 0.016 | 0.006 | 2.670 | 0.012 | 0.145 |
| SELF-CONCEPT | ← | BMI | −17.183 | 3.421 | −5.022 | *** | −0.277 |
| EC | ← | TC | −0.164 | 0.023 | −7.292 | *** | −0.357 |

Note 1: TC, task-oriented climate; E/I, effort/improvement; CL, cooperative learning; IR, important role; EC, ego-oriented climate; MR, rivalry between members; UR, unequal recognition; PM, punishment of mistakes; MD, Mediterranean diet; BMI, body mass index; SELF-CONCEPT, self-concept; Note 2: R.W., Regression Weights; S.R.W., Standardised Regression Weights; E.E., Estimation Error; C.R., Critical Ratio; Note 3: $p < 0.001$***.

When analyzing the influence of the indicators, statistically significant associations at the level of $p < 0.001$ can be seen, with all of these associations being positive. In the case of the TC, the E/I is the indicator that shows the strongest relation coefficient (r = 0.877), followed by CL (r = 0.726) and the IR (r = 0.725). For the EC, the strongest association is given with UR (r = 0.872), followed by PM (r = 0.759) and MR (r = 0.616). These values show a high strength in all of the regression weights, except for the category of MR, with a medium strength.

In the same way, a significant association is observed ($p < 0.01$) for the relationship between the TC and BMI, this being negative and weak (r = −0.153). The relationship between the TC and MD is significant ($p < 0.001$), being positive (r = −0.222) and showing a low regression weight. Furthermore, statistically significant associations at a level of $p < 0.001$ are found between the TC and self-concept, suggesting a positive association (r = 0.571) with a medium strength of the relationship.

The relationship between the EC and BMI reveals a statistically significant association at the level of $p < 0.05$, showing a positive relationship and a low regression weight (r = 0.081). In addition, a relationship was found between the EC and MD ($p < 0.05$), with this association being negative with a low regression weight (r = −0.111). With regards to the association between the EC and self-concept, this was also found to be significant at the level of $p < 0.05$, with this relationship being negative with a low strength for the regression weight (r = −0.079).

An analysis of the relationship between the MD and BMI revealed a negative and indirect association (r = −0.174; $p < 0.001$) of low relation strength. Alternatively, the relationship examined between the MD and self-concept was found to be positive (r = 0.145; $p < 0.01$). Finally, the association between BMI and self-concept was negative and indirect (r = −0.277), with a significance level of $p < 0.001$ and a low strength for the regression weight.

## 4. Discussion

The present research developed a multi-group structural equation analysis with the objective of contrasting the associations produced between the motivational climate towards sport, adherence to a Mediterranean diet, self-concept, and body mass index. The path model developed demonstrated excellent fit indices, suggesting that the explanatory model validly explained the existing relationships between motivational factors, diet, body mass index, and self-concept in the present sample of school children of both sexes, to the same extent as other models produced in previous studies [35–39].

In terms of the motivational climate, the path model showed a significant and negative relationship between the ego and the task-oriented climate in both sexes, with the association being stronger and more differentiated amongst girls. The present theoretical model confirmed the presence of an inverse association between the task-oriented climate and the ego-oriented climate, with a low strength in the regression weights for the boys and girls. This reveals that school children who report an elevated task-oriented motivation, enabling them to focus on effort and overcoming personal challenges, tend to display low levels of ego orientation, promoting intergroup rivalry, and vice versa [40–42]. The inverse relationship between both dimensions of the motivational climate was stronger amongst females, with males reporting higher scores for the ego-oriented climate and lower scores for the task-oriented climate. This may be explained by the finding that females tend to focus on intrinsic goals, whereas males tend to favor extrinsic goals [43,44].

In consideration of the contribution of the categories of the task climate, the indicator showing the greatest relation strength, in both males and females, was effort/improvement, with a low strength in the regression weights for all of the indicators of the task and ego climate except member rivalry in both sexes. The indicator exercising the least influence for females was an important role and cooperative learning in the case of males. With regards to the ego climate, outcomes relating to the influence of indicators showed a similar pattern, with unequal recognition being the indicator exercising the greatest influence within both males and females, followed by the punishment of mistakes and rivalry between members. Effort/improvement represents the factor most strongly characterized by high levels of intrinsic motivation, as the individual is driven by personal progress and leaves aside the attainment of extrinsic goals [45]. Boys attributed less importance to cooperative learning, with an important role being the least important factor for girls. From this, it can be concluded that girls value the process of learning and collaborating with other group members to a greater extent, while boys value the attainment of an important role and standing out from others due to their own personal abilities more highly [46,47].

The association between the task climate and body mass index was negative and indirect in both groups, with a greater relation strength being evident amongst girls. The relationship between the ego climate and body mass index was positive and direct in both groups, but it demonstrated a slightly stronger relation amongst boys in this case. When the task orientation increases, the body mass index decreases, and when the ego orientation increases, the body mass index increases. This can be explained by the fact that more self-determined or intrinsic motivations augment factors that are related to healthy lifestyles, while more extrinsic or less self-determined motivations are related to less healthy behaviors [48,49]. Individuals who focus on the task typically take better care of their body, as they tend to prioritize factors related to effort and improvement through personal training. On the other hand, individuals oriented more towards the ego are enthused by the attainment of external rewards [50], attributing less importance to personal effort. For this reason, a negative relationship between both dimensions of the motivational climate is found, which increases in adolescence.

When the relationship between the task climate and the Mediterranean diet is examined, a positive direct association is identified in both boys and girls, with a stronger relation being ascertained in girls. In contrast, the ego-oriented climate is negatively associated with Mediterranean diet adherence in both groups, with stronger relations found amongst boys. From this data, it can be concluded that a direct relationship exists between the task climate and Mediterranean diet adherence, while an inverse relationship exists in the case of the ego climate. In line with comments in the previous paragraph,

intrinsic motivations are related to the development of healthy habits and optimal nutritional patterns, while the ego climate encourages less healthy habits, which will also affect the nutritional profile of individuals [51,52]. During this life stage of school children, dietary patterns depend, to a large degree, on parental influence, though a decrease in this influence begins to emerge [53]. Differences relating to Mediterranean diet adherence between the sexes can be explained by the greater task-oriented motivation reported by girls and the greater ego-orientation reported by boys, as was the case with regards to the body mass index.

An analysis of the relationship between the task climate and self-concept of these school children found a positive direct association in both groups, with a stronger relation identified amongst girls. Furthermore, the association between the ego-oriented climate and self-concept was negative and indirect, with a stronger association amongst boys. Motivational orientations towards the task are allied to the intrinsic motivations which encourage overcoming personal challenges, effort, and the belief that improvement is possible, while orientation towards the ego relates with factors connected to the pure demonstration of ability and overcoming the opponent. Individuals who concentrate on personal improvement through training will also have higher levels of self-concept, as they are less likely than ego-oriented individuals to experience a fear of failure. In contrast, those who are driven purely to demonstrate their ability as a mechanism of gaining external rewards, risk suffering from a fear of failure and a diminished self-concept [54]. The perception held by an individual about themselves is, in turn, affected by their response to different situations arising during daily life. Individuals who are more task-oriented present better levels of self-concept than those who are more ego-oriented, due to their need to stand out and the concomitant fear of failure [47]. With regards to differences according to sex, the present study suggests that females are more inclined to focus on overcoming personal challenges and, as a result, report a better self-concept. Males give greater importance to competitive factors, which produces a self-concept that is more dependent on concrete achievements and failures [55].

A negative relationship was found between Mediterranean diet adherence and body mass index, with the strength of the relation being similar between boys and girls. This result clearly demonstrates the benefits of the Mediterranean diet, which has been previously related to a better state of general health, a lower body mass index, and a reduced risk of suffering from cardiovascular diseases [56]. Given that the nutritional patterns characteristic of the Mediterranean diet are associated with reduced obesity, reduced excess weight, and better general health, its consumption will markedly affect the self-concept, particularly the physical self-concept, reported by school children [57,58].

Finally, when the relationship between body mass index and self-concept is considered, a negative and indirect relationship is seen, with a slightly stronger relation amongst girls. The association between body mass index and self-concept can be largely explained by the strong impact of body image on the physical dimension of self-concept, which, in turn, has a strong impact on the social dimension of self-concept [59,60]. The differences presented between sexes, in which males showed a stronger relation, can be explained by the fact that boys of the age examined in the present study tend to engage in physical activity competitively against their peers. In this context, the maintenance of a good state of health and lower body mass index will result in a better perception of their physical self-concept, which will improve the development of physical activities or sports at the same time [61]. On the other hand, girls focus less on the development of competition, which explains the weaker relation discussed here [62].

## 5. Implications for Practice

The following practical implications from the findings of the present study are proposed. Firstly, dietary patterns characteristic of the Mediterranean diet must be promoted to school children alongside a task-oriented motivational climate, due to their positive influence with regards to body mass index, which also positively influences general self-concept. The findings of the present study should be considered in light of a number of limitations. Amongst them, the descriptive cross-sectional research

design is highlighted as a weakness by preventing the identification of causal relationships. Moreover, it is important to highlight that the results must be analyzed with caution, due to the fact that some associations between the variables analyzed showed low regression weights. Furthermore, it will be of interest to future studies to include a greater number of psychological variables, with the aim of analyzing their association with the factors included in the present study.

## 6. Conclusions

This research establishes two main conclusions:

- The task-oriented motivational climate is more prevalent than the ego-oriented climate, showing an inverse relationship between both variables which is stronger in girls. The category of the task climate that exercised the greatest influence was effort or improvement, while unequal recognition was the most influential dimension of the ego climate in both sexes.

- An inverse relationship was found to exist between the task climate and body mass index, and a direct relationship was found between the ego climate and body mass index. Mediterranean diet adherence was directly related to the task climate, while an ego climate was negatively associated. A positive relationship exists between the task climate and the self-concept of school children, while the relationship with the ego climate is negative, suggesting that a greater orientation towards the task improves self-concept. Greater adherence to the Mediterranean diet is related to a lower body mass index. Finally, a negative relationship is found between self-concept and body mass index, showing that a lower body mass index improves self-concept.

**Author Contributions:** M.C.-S., F.Z.-O., and R.C.-C. conceived the hypothesis of this study. M.C.-S., F.Z.-O., R.C.-C., and E.G.-M. participated in the data collection. M.C.-S. and R.C.-C. analyzed the data. All authors contributed to the data interpretation of the statistical analysis. M.C.-S., F.Z.-O., and R.C.-C. wrote the paper. All authors read and approved the final manuscript.

**Funding:** This research received no external funding.

**Conflicts of Interest:** The authors declare no conflict of interest.

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
