# Peer review of "Motivational Climate towards the Practice of Physical Activity, Self-Concept, and Healthy Factors in the School Environment"

_sustainability, doi:10.3390/su11040999_

Round 1
Reviewer 1 Report
The manuscript "Motivational climate towards the practice of physical activity, self-concept and healthy factors in the school environment" investigates motivation and relationship between (sociodemographic) variables regarding health and mediterranean diet.
Introduction:
Authors presented a interesting bibliography regarding motivation and its influences in physical activities as also a brief but well understandable summary about mediterranean diet and its positives influences in ones health (literature).
Material and Methods:
Experimental design and sample size are well explained and defined, sample ist balanced regarding sex distribution but no information about the type of school (public / private) was presented nor the distribution of schoolchildren per school.
The motivational climate questionnaire (PMCSQ-2) delivered with this sample good internal reliability indexes having only one marginal value for rivalry. The Mediterranean diet quality index (KIDMED) questionnaire as also the Self-concept form-5 (AF-5) had also a (very) good intern reliability index.
Line 152: "a total of 52 questionnaires were excluded" do the authors mean that 52 observations were exclueded? How did they deal with missing values in some variables? Which procedure was chosen to handle them?
Statistical Analysis
Authors have decided for a multi-group structural model grouping data according to sex but they didn't present any information that confirms this necessity of generate two models, why is the reason for such choice? Did they pre-analyzed the data and found any statistical significance between genders that would lead them to this path?
Lines 171 -172 "Bidirectional arrows show the relationship between the latent variables and also represent regression coefficients." Where are the bidirectional arrows which represent regression coefficient? Aren't they the onedirectional as described?
No information about the explorative factor analysis regarding the definition of both latent variables TC and EC was presented, how do the variables relate here? SEM is a confirmative method.
Results:
Authors must present p-values in Tables 1 and 2, only highlighting them with * is not informative enough. In Figure 2 and 3 authors report values as being correlation altough they are presented as onedirectional arrows, this must be corrected to avoid misunderstanding. They also report some statistical significant correlation that aren't strong to be considered, for example between ego climate and body mass index the correlation is r=0,09 which is very weak. They didn't present a reference for the correlation values used - what do they consider as weak or strong correlation?
Discussion:
Authors recall the objectives of the study and results from statistical modeling but as pointed above the model / model results presentation msut be improved. They argue on results that somehow are statistical significant (regarding p-values which is a very critical and sensitive statistic measure) as if only theses p-values and not the raw correlation values are important.
They presented results for both separeted models (female and male) but didn't present an overall model which could be as informative as both of them.
Final comments:
Statistical models (explorative factor analysis and sem) must be improved, statistics are missing and information in table and figures are confuse.
Author Response
Dear the editor and reviewers,
We would like to express our gratitude for the time taken to review this manuscript and for the comments made, which we believe to be critical for producing rigorous and quality research. We have detailed below the changes made in the original article: “Motivational climate towards the practice of physical activity, self-concept and healthy factors in the school environment” (sustainability-439823).
Modifications have been made in the original manuscript following the reviewers’ comments. For each modification we have written: the original comment as written by the reviewer in addition to the page and line number; and the change made in response to that comment. Changes have been made using the tool “Track changes” enabling editor and reviewers to identify modifications easily.
Reviewer 1
Comment 1:
Material and Methods:
Experimental design and sample size are well explained and defined, sample ist balanced regarding sex distribution but no information about the type of school (public / private) was presented nor the distribution of schoolchildren per school.
The motivational climate questionnaire (PMCSQ-2) delivered with this sample good internal reliability indexes having only one marginal value for rivalry. The Mediterranean diet quality index (KIDMED) questionnaire as also the Self-concept form-5 (AF-5) had also a (very) good intern reliability index.
Line 152: "a total of 52 questionnaires were excluded" do the authors mean that 52 observations were exclueded? How did they deal with missing values in some variables? Which procedure was chosen to handle them?
Response 1:
Thanks for this indication. The students who have participated in this research come from all public schools, so in the sample section and in the summary it has been indicated that the students analyzed come from public schools. Regarding the distribution of students by educational centre, due to the fact that they are under-age students, when permission was requested from educational centres to collect data, most of those responsible for the schools told us that they preferred to by anonymity and that the name of the educational centre does not appear in the investigations, therefore the distribution of students by educational centre is not included. However, if the reviewer deems it convenient, the distribution of students by educational centre can be included, naming the centres as centre 1, centre 2, centre 3.
Thank you very much for the appreciation regarding the reliability of the instruments used in the investigation. You are right, and we have observed that in the majority of investigations carried out on the motivational climate, the reliability regarding the rivalry category among the members of the group is usually lower.
We appreciate your third comment, since this will clarify the process. The clarification has been included indicating that the original sample consisted of a total of 786 schoolchildren, but it was decided to eliminate 52 questionnaires because when they were revised to create the database, it was found that there were some items that were not filled. Thus, a total of 52 questionnaires were they were not valid.
Comment 2:
Statistical Analysis
Authors have decided for a multi-group structural model grouping data according to sex but they didn't present any information that confirms this necessity of generate two models, why is the reason for such choice? Did they pre-analyzed the data and found any statistical significance between genders that would lead them to this path?
Lines 171 -172 "Bidirectional arrows show the relationship between the latent variables and also represent regression coefficients." Where are the bidirectional arrows which represent regression coefficient? Aren't they the onedirectional as described?
No information about the explorative factor analysis regarding the definition of both latent variables TC and EC was presented, how do the variables relate here? SEM is a confirmative method.
Response 2:
Thanks for this indication. In order to define more specifically the relationships of the constructs that make up the route model, multigroup analysis was carried out, since several studies show that the motivational climate varies according to sex, and the motivational climate is our exogenous variable (Zurita, F., Castro, M., Chacón, R., Cachón, J., Cofré, C., Knox, E., Muros, J. J. (2018). Analysis of the Psychometric Properties of Perceived Motivational Climate in Sport Questionnaire and Its Relationship to Physical Activity and Gender Using Structural Equation Modelling, Sustainability, 10(3), 632-642.).
We have done previous studies that we have done and show the differences between sexes, such as:
-Castro-Sánchez, M., Zurita-Ortega, F., Martínez-Martínez, A., Chacón-Cuberos, R., & Espejo-Garcés, T. (2016). Motivational climate of adolescents and their relationship to gender, physical activity, sport, federated sport and physical activity family. RICYDE. Revista Internacional de Ciencias del Deporte, 12(45), 262-277. doi: 10.5232/ricyde.
- Castro-Sánchez, M., Chacón-Cuberos, R., Ubago-Jiménez, J., Zafra-Santos, E., & Zurita-Ortega, F. (2018). An explanatory model for the relationship between motivation in sport, victimization, and video game use in schoolchildren. International journal of environmental research and public health, 15(9), 1866.
There is a typographical error in the article, and there is only one bidirectional arrow. Thank you very much for warning us of this error, it has already been corrected.
-Marsh (2007) indica que la relacion entre las exogenas (elipse) se representan mediante flechas bidireccionales, mientras que el resto de relaciones con flechas unidireccionales, ya que son variables endógenas o implican a variables endógenas:
-Marsh, H.W. Handbook of Sport Psychology. Third Edition. New Jersey, USA, 2007.
Thank you very much for all your comments and contributions. Exploratory factor analysis of this same questionnaire have been carried out in different samples, such as schoolchildren, adolescents, and university students:
-Walling, M. D., Duda, J. L., & Chi, L. (1993). The perceived motivational climate in sport questionnaire: Construct and predictive validity. Journal of Sport and Exercise Psychology, 15(2), 172-183.
- Balaguer, I., Duda, J. L., Atienza, F. L., & Mayo, C. (2002). Situational and dispositional goals as predictors of perceptions of individual and team improvement, satisfaction and coach ratings among elite female handball teams. Psychology of Sport and Exercise, 3(4), 293-308.
- Soini, M., Liukkonen, J., Watt, A., Yli-Piipari, S., & Jaakkola, T. (2014). Factorial validity and internal consistency of the motivational climate in physical education scale. Journal of sports science & medicine, 13(1), 137.
- Flores, J., Salguero, A., & Márquez, S. (2008). Goal orientations and perceptions of the motivational climate in physical education classes among Colombian students. Teaching and Teacher education, 24(6), 1441-1449.
So we deduce that this instrument is valid to measure the motivational climate in our sample. On the other hand, the SEM is not a confirmatory analysis as it indicates, since we do not check the loading of each item to its dimension, but we make a model of routes with different constructs obtained from different instruments (motivational climate, adherence to the Mediterranean diet and self-concept). That is why we believe that the analysis is adequate, we cite similar studies:
- Jaakkola, T., Ntoumanis, N., & Liukkonen, J. (2016). Motivational climate, goal orientation, perceived sport ability, and enjoyment within Finnish junior ice hockey players. Scandinavian journal of medicine & science in sports, 26(1), 109-115.
- Ruiz, M. C., Haapanen, S., Tolvanen, A., Robazza, C., & Duda, J. L. (2017). Predicting athletes’ functional and dysfunctional emotions: The role of the motivational climate and motivation regulations. Journal of sports sciences, 35(16), 1598-1606.
- Baena-Extremera, A., Gómez-López, M., Granero-Gallegos, A., & Ortiz-Camacho, M. D. M. (2015). Predicting satisfaction in physical education from motivational climate and self-determined motivation. Journal of Teaching in Physical Education, 34(2), 210-224.
Comment 3:
Results:
Authors must present p-values in Tables 1 and 2, only highlighting them with * is not informative enough. In Figure 2 and 3 authors report values as being correlation altough they are presented as one directional arrows, this must be corrected to avoid misunderstanding. They also report some statistical significant correlation that aren't strong to be considered, for example between ego climate and body mass index the correlation is r=0,09 which is very weak. They didn't present a reference for the correlation values used - what do they consider as weak or strong correlation?
Response 3:
Thank you very much for this suggestion of improvement. Specifically, in the case of the *** values (p <0.001), the Amos program directly indicates the asterisks, it does not indicate the numerical value by the number of decimals they contain. On the contrary, for an asterisk and for two the values of p have been included.
Unidirectional arrows are used because the Amos program oblige us to use these arrows between endogenous variables. A regression weight is obtained that can be positive or negative, and this indicates the relationship between said variables, therefore, the unidirectional arrows must be used. However, we appreciate your indication, since we have detected that the error is found in the writing of the manuscript, since the word correlation is used. In this case the word correlation has been replaced by relation or association.
Thank you for indicating this. In some relationships the strength of the regression weights have not been indicated, and we have proceeded to include it. We consider them to be 0-0.3 low, from 0.31 to 0.60 average and from 0.61 to 0.99 high. As the regression coefficient oscillates between 0 and 1, a division of it has been considered in tertiles, as indicated by Marsh (2007).
-Marsh, H.W. Handbook of Sport Psychology. Third Edition. New Jersey, USA, 2007.
Comment 4:
Discussion:
Authors recall the objectives of the study and results from statistical modeling but as pointed above the model / model results presentation msut be improved. They argue on results that somehow are statistical significant (regarding p-values which is a very critical and sensitive statistic measure) as if only theses p-values and not the raw correlation values are important.
They presented results for both separeted models (female and male) but didn't present an overall model which could be as informative as both of them.
Response 4:
Thanks for this indication. We have proceeded to point out that the regression weights are low, including the description about the differences according to both sexes. In addition, in the limitations of the study this information has been included.
The multigroup SEM has been used because the interest of this research was to detect the differences between sexes, in fact, better fit indexes were obtained for the multigroup model than for the general model.
Comment 5:
Final comments:
Statistical models (explorative factor analysis and sem) must be improved, statistics are missing and information in table and figures are confuse.
Response 5:
Thanks for this suggestion of improvement. Thanks for this indication. However, the authors believe that all the necessary steps have been taken at the statistical level to carry out a study of these characteristics, since:
1-Factor analysis is not performed, since these instruments are previously validated in the same sample. In fact, there are similar studies that use SEM and do not perform it.
2-We propose the theoretical model for which we obtain good adjustment indices (CFI, NFI, IFI, RMSEA).
3-As good adjustment rates are obtained, we proceed to check the relationships between variables, which are statistically significant. In fact, the way of presenting the statistics is analogous to other studies published in quality journals, such as:
- Zurita Ortega, F., Castro Sánchez, M., González, Á., Ignacio, J., Rodríguez Fernández, S., & Pérez Cortés, A. J. (2016). Autoconcepto, Actividad física y Familia: Análisis de un modelo de ecuaciones estructurales. Revista de psicología del deporte, 25(1), 0097-104.
- Cuberos, R. C., Ortega, F. Z., Sánchez, M. C., Garcés, T. E., Martínez, A. M., & Ruiz, G. R. R. (2017). Relación entre autoconcepto, consumo de sustancias y uso problemático de videojuegos en universitarios: un modelo de ecuaciones estructurales. Adicciones.
- Martínez, A. M., Sánchez, M. C., Fernández, S. R., Ortega, F. Z., Cuberos, R. C., & Garcés, T. E. (2018). Conducta violenta, victimización, autoestima y actividad física de adolescentes españoles en función del lugar de residencia: un modelo de ecuaciones estructurales. Revista de Psicología Social, 33(1), 125-141.
- Zurita-Ortega, F., Castro-Sánchez, M., Rodríguez-Fernández, S., Cofré-Boladós, C., Chacón-Cuberos, R., Martínez-Martínez, A., & Muros-Molina, J. J. (2017). Actividad física, obesidad y autoestima en escolares chilenos: Análisis mediante ecuaciones estructurales. Revista médica de Chile, 145(3), 299-308.
- Baena-Extremera, A., Granero-Gallegos, A., Ponce-de-León-Elizondo, A., Sanz-Arazuri, E., Valdemoros-San-Emeterio, M. D. L., & Martínez-Molina, M. (2016). Factores psicológicos relacionados con las clases de educación física como predictores de la intención de la práctica de actividad física en el tiempo libre en estudiantes. Ciência & Saúde Coletiva, 21, 1105-1112.
For all this, we believe that there is no missing data and that the way to present the statistics is consistent and according to the scientific method.
Reviewer 2 Report
The idea of study is interesting and complex because it tried to made connection between Motivational climates, Mediterranean diet quality index, body mass index and self-concept and healthy factors in the school environment.
Recommendations:
Introduction – Need to be extending with more relevant information focused on the particularities of age sample of study (10-12 years old).
Please highlight the novelty of your study according with yours and previous researches. Part of authors approached a big part of the topic of manuscript in other articles.
I recommend to extend Discussion with more relevant and recent information to sustain your findings of research.
I recommend adding one or two specific conclusions related to your research.
Author Response
Dear the editor and reviewers,
We would like to express our gratitude for the time taken to review this manuscript and for the comments made, which we believe to be critical for producing rigorous and quality research. We have detailed below the changes made in the original article: “Motivational climate towards the practice of physical activity, self-concept and healthy factors in the school environment” (sustainability-439823).
Modifications have been made in the original manuscript following the reviewers’ comments. For each modification we have written: the original comment as written by the reviewer in addition to the page and line number; and the change made in response to that comment. Changes have been made using the tool “Track changes” enabling editor and reviewers to identify modifications easily.
Reviewer 2
Comment 1:
Introduction – Need to be extending with more relevant information focused on the particularities of age sample of study (10-12 years old).
Response 1:
We appreciate your indications as they improve the quality of the article. After analyzing the psychological variables and the factors related to health, the characteristics of schoolchildren aged 10 to 12 years have been developed in a paragraph, highlighting the importance of conducting research in this type of sample.
Comment 2:
Please highlight the novelty of your study according with yours and previous researches. Part of authors approached a big part of the topic of manuscript in other articles.
Response 2:
We appreciate this indication as it will improve the quality of the manuscript. A paragraph has been included in the introduction to talk about the novelty of this research and the difference between those made previously, highlighting that the association between psychological factors (motivational climate and self-concept) and variables related to health is analyzed. (adherence to the Mediterranean diet and body mass index).
Comment 3:
I recommend to extend Discussion with more relevant and recent information to sustain your findings of research.
Response 3:
Thanks for this indication. In the discussion, 25 bibliographical references of the years 2016, 2017 and 2018 have been used, finding a total of one bibliographic reference of the year 2016, fifteen of the year 2017 and nine of the year 2018. The majority of the bibliographic references used are extracted from journal of impact published in the Web of Science.
Comment 4:
I recommend adding one or two specific conclusions related to your research.
Response 4:
Thanks for this suggestion as it improves the quality of the manuscript. We proceeded to structure the conclusions of the investigation in two specific conclusions.
Round 2
Reviewer 1 Report
Authors have explained some unclear Points and changed the text in order to better explain the procedures and methods adopted.
After some explanations the questions were clarified and are now inderstandable.
Authors improve the text making it more didatic and easy to read for Readers.
Theme is interesting, well done.
Reviewer 2 Report
The authors improved the manuscript according with the recommendations.